# Closing the gender gap in competitiveness through priming

Loukas Balafoutas [1], Helena Fornwagner[1] & Matthias Sutter [1,2,3]

Men have been observed to have a greater willingness to compete compared to women, and it is possible that this contributes to gender differences in wages and career advancement. Policy interventions such as quotas are sometimes used to remedy this but these may cause unintended side-effects. Here, we present experimental evidence that a simple and practically costless tool—priming subjects with power—can close the gender gap in competitiveness. While in a neutral as well as in a low-power priming situation men are much more likely than women to choose competition, this gap vanishes when subjects are primed with a high-power situation. We show that priming with high power makes competition entry decisions more realistic and also that it reduces the level of risk tolerance among male participants, which can help explain why it leads to a closing down of the gender gap in competitiveness.

[1] Department of Public Finance, University of Innsbruck, Universitätsstraße 15, 6020 Innsbruck, Austria. [2] Max Planck Institute for Research on Collective Goods, Kurt-Schumacher-Straße 10, 53113 Bonn, Germany. [3] Department of Economics, University of Cologne, Albertus-Magnus-Platz, 50923 Köln, Germany. Correspondence and requests for materials should be addressed to L.B. (email: loukas.balafoutas@uibk.ac.at)

Modern societies are still characterized by large differences between men and women, covering many different dimensions like status in society, access to resources, or success on labor markets. The persistent gender differences on labor markets perpetuate inequalities between men and women, for which reason academic research has been investigating the causes and consequences of gender differences for many years. Recently, a significant part of the large differences in labor market outcomes of men and women has been attributed to gender differences in performance under competitive pressure and in the willingness to compete[1–7]. Typically, women are observed to shy away from competition more often than men, which may explain poorer career prospects of women with respect to wages and promotions. As a consequence, policy interventions in the form of affirmative action programs have received much support both in the academic debate and among the general public. Various interventions have been found to promote gender-balanced outcomes in labor markets, including quotas[8–10], preferential treatment of women[9], prizes benefiting one's offspring[11], or incentives based on co-operation[12]. Nevertheless, such policies typically require (costly) institutional changes, are subject to skepticism and opposition, or hard to implement on practical grounds. There is even growing evidence that affirmative action programs can backfire against women[13,14].

For these reasons, we consider it as important to examine complementary, less intrusive and inexpensive, ways to mitigate or even offset the gender differences in competitiveness. Companies and their human resource departments may benefit from such alternatives because they have an interest in selecting the best candidates for a job, irrespective of gender, which requires the best candidates to accept the challenges of competition and feel comfortable in such situations. The latter conditions are more easily met when subjects feel in charge and are convinced that they can—metaphorically speaking—master their fate. This line of reasoning suggests that feelings of being in control may help reduce the gender gap in competitiveness. Therefore, we present a novel design where we combine the psychological tool of priming with power with an economic experiment on competitiveness. Our hypothesis is that when a feeling of power is activated, this might create in women a feeling of being in control to achieve one's goals and hence could close the gender gap in the willingness to compete.

Priming has been used extensively in psychological research. This method refers to the temporal internal activation of response tendencies by, e.g., making specific identities salient[15,16], activating particular feelings[17], and putting individuals in certain mind-sets[18]. Applied in different contexts and on subject groups with a variety of socio-economic backgrounds[19,20], priming has been shown to successfully change how people think and behave. In this study we use a priming method to activate a feeling of power, which has been reported to let subjects (both men and women) work more persistently on complex tasks[21], prioritize more successfully on important goals[17,22,23], take riskier choices[24], and increase the level of moral hypocrisy or anti-social behavior[17,25]. This paper is the first that brings together power priming and an incentivized economic experiment[26–28].

We find that in a neutral as well as in a low-power priming situation men are much more likely than women to choose competition. However, this gap vanishes when subjects are primed with a high-power situation. We propose mechanisms that can explain this result, showing that priming with power affects risk aversion and the extent to which competition decisions are realistic and well calibrated. Overall, our findings suggest that a simple and inexpensive intervention based on priming with power can contribute to a gender-balanced pattern in competitive behavior.

## Results

**Power priming and validity check.** At the beginning of an experimental session, participants were randomly allocated to one of three treatments: a neutral treatment, a high-power or a low-power priming treatment. For the power priming treatments, we used a writing task[17] that has been reported to reliably manipulate one's sense of power. Participants assigned to the high-power priming (HIGH) condition were asked to recall and write about a personal situation in which they had control over another individual or individuals, while those assigned to the low-power priming (LOW) condition were instructed to write about a personal situation in which someone else had control over them (see Supplementary Methods for details). A validity check after the completion of the priming task revealed that our priming worked well. When asked to state on a scale from 1 (absolutely powerless) to 9 (absolutely powerful) how much they felt in power in their described situation, subjects reported significantly higher numbers on average in HIGH (6.22) than in LOW (3.41; Mann–Whitney $U$, $z = -9.39$, $P < 0.001$, $n = 256$, see Supplementary Table 1 for more details). Disaggregating by gender, we document no differences between men and women in the reported scale in LOW (3.43 vs. 3.38; Mann–Whitney $U$, $z = -0.06$, $P = 0.952$, $n = 122$) and HIGH (6.16 vs. 6.29; Mann–Whitney $U$, $z = -0.02$, $P = 0.898$, $n = 134$). Finally, in treatment NEUTRAL, which serves as a baseline, participants were not primed and did not complete the writing task or report a scale.

Before running the competition experiment, participants were asked to fill out a mood elicitation questionnaire (MDMQ[29]) to measure their current emotional state. The MDMQ consisted of questions related to 12 mood dimensions, which were split into three categories: Good-Bad-Mood (GB), Awake-Tired-Mood (AT), and Calm-Nervous-Mood (CN). The participants were asked to evaluate each stated mood on a scale from 1 (not at all) to 5 (absolutely). Mean responses by treatment are shown in Supplementary Table 2. We also collected some additional socio-demographic information of the participants, probed them for suspicion using a funnel questionnaire, and debriefed them[30].

**The competition experiment.** We randomly assigned participants into groups of two men and two women (four persons in total per group). Other than this gender composition, participants received no further information regarding their potential competitors. All groups went through three stages that were programmed with z-Tree[31]. The experimental task in each stage was to add up as many two-digit numbers as possible within two minutes[5]. In Stage 1, called "piece rate", each participant received €1 for each correct calculation. In Stage 2, "tournament", group members were in a competition against each other. Only the member who solved the highest number of calculations correctly was paid €4 per correct calculation, all others received no payment. In Stage 3, "choice", participants could choose before solving the calculations whether they wanted to do so under the piece rate (as in Stage 1) or the tournament payment scheme (as in Stage 2). If a participant chose the tournament scheme, his or her performance in this stage was then compared to the other group members' performance in Stage 2 (which transforms the choice in Stage 3 into an individual decision making task, thus avoiding that expectations about other members' entry decisions might affect one's choice). If a participant's performance in Stage 3 was higher than the ones of all other group members in Stage 2, the participant received €2 for every correctly solved calculation, otherwise zero. Ties were broken randomly.

These incentives differ with respect to most of the previous literature, since entering the competition in our experiment

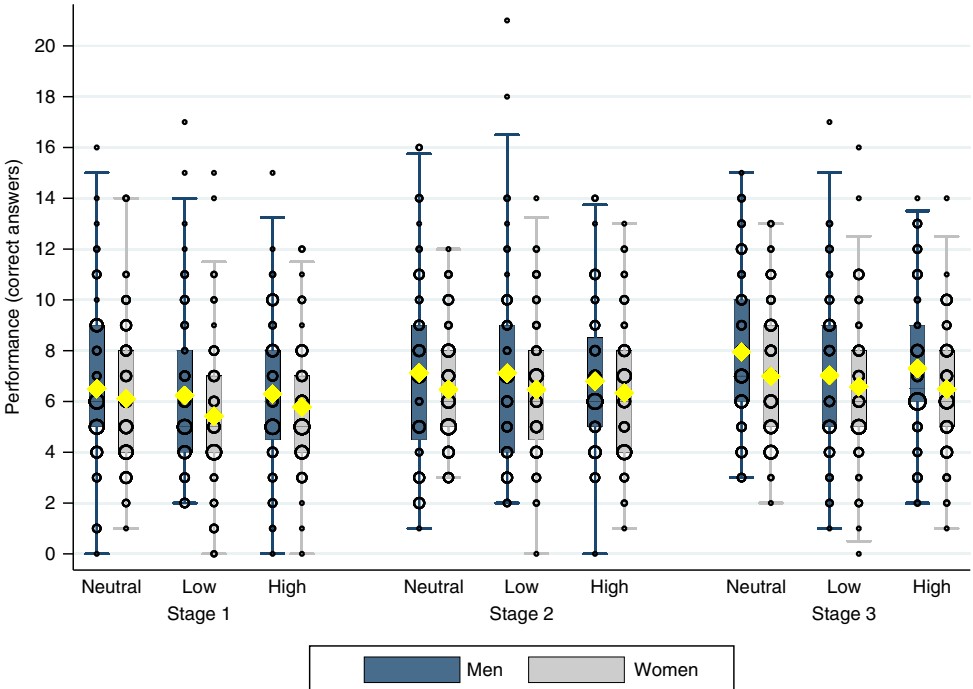

**Fig. 1** Performance in the real effort task, by gender, stage, and priming condition ($n = 401$). Box plots show the mean (indicated by yellow diamond signs), the 25th and 75th percentiles, Tukey whiskers (median ± 1.5 times interquartile range), and individual data points. Larger dots indicate a higher number of participants with the corresponding performance

yields, on average, lower expected payoffs than choosing the piece rate in Stage 3. Actually, it is quite often the case that tournaments pay less than piece rates on average. Recent evidence suggests that cost-minimizing firms prefer to choose the lowest tournament prizes possible, since these would still be enough to attract good applicants, in particular, when they are overconfident and have a much higher subjective probability of winning the tournament than the objective probability justifies. Such conditions yield lower expected payoffs from the tournament than from a piece rate[32], for which reason we consider our incentives in Stage 3 to be a good representation of many real-world situations. Moreover, if the gender gap persists under these conditions, this would show that previous findings are robust to a change in incentives and also that our results are comparable to the findings in the existing literature[33].

While working on the tasks in the three stages, participants received feedback on whether an entry was right or wrong, and at the end of each stage they were informed about the total number of right answers in the respective stage. Yet, they did not receive any information about the other group members' performance until the end of the experiment. We elicited also participants' beliefs about how their performance ranked within their session (for Stage 1) or group (for Stage 2), and correct beliefs were rewarded with €1 each (see Supplementary Methods for details). At the end of the experiment, one of the three stages was randomly chosen for payment. Following the previous literature[5], we also controlled for risk preferences by administering a simple investment game[34] after Stage 3 (see Supplementary Methods and Supplementary Figure 1 for details), because competition entry decisions are likely to depend upon risk attitudes.

We had 424 participants in our study (212 men, 212 women). In the following presentation of results, we drop 23 participants because in a debriefing questionnaire they indicated suspicion of being primed. This leaves us with data for 401 participants (see Supplementary Table 3 for their gender composition).

**Performance**. Figure 1 presents the performances of women and men across all stages, split up by priming condition. In line with most previous experimental results, gender differences in performance are not significantly different in any of the stages (Mann–Whitney $U$ for nine pairwise comparisons, $P > 0.176$), with the partial exception of Stage 3 in NEUTRAL, where men perform slightly better than women (Mann–Whitney $U$, $z = 1.77$, $P = 0.077$, $n = 144$). Moreover, there are no differences in male or female performance across priming conditions (six Kruskal–Wallis tests by stage and gender, $P > 0.231$).

**Competition entry decisions**. Figure 2 shows the relative frequency with which participants in each treatment chose to compete in Stage 3. We find that in the neutral condition men competed around three times as often as women (40.3% vs. 13.9%; $\chi^2(1) = 12.69$, $P < 0.001$, $n = 144$). This gender difference persists—albeit at slightly smaller magnitude—in the low-power condition LOW, with male and female entry rates of 38.1% and 20%, respectively ($\chi^2(1) = 4.86$, $P = 0.027$, $n = 123$). However, in the HIGH condition this gender gap is drastically reduced and becomes insignificant due to the fact that women increase and men decrease their entry rates relative to NEUTRAL. Entry rates in HIGH are 27.9% for men and 19.7% for women ($\chi^2(1) = 1.25$, $P = 0.263$, $n = 134$). Although the decrease (increase) in male (female) entry rates in HIGH compared to NEUTRAL are not significant ($\chi^2(1) = 2.36$, $P = 0.12$, $n = 140$ for men; $\chi^2(1) = 0.84$, $P = 0.36$, $n = 138$ for women), the resulting difference-in-differences corresponding to the effect of the high-power prime on the gender gap is significant, as the following regression analysis reveals. In Supplementary Table 4 we confirm the non-parametric results through Probit regressions. In the first specification (column 1), the gender gap in competition entry is found to be significant in treatments NEUTRAL (as captured by the coefficient of −0.84 on female, $z = -3.54$, $P < 0.001$, $n = 401$) and LOW (as captured by the joint coefficient female + female ×

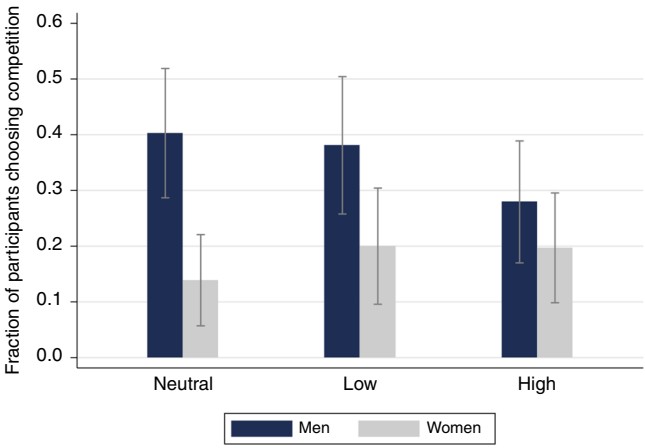

**Fig. 2** Proportion of participants choosing competition in Stage 3 ($n = 401$). The bars show, for each priming condition, the proportion of participants (between 0 and 1) who chose the tournament compensation rather than the piece rate. Error bars, mean ± S.D.

LOW $= -0.90$, $\chi^2(1) = 4.85$, $P = 0.028$, $n = 401$), but insignificant in HIGH (coefficient of female + female × HIGH $= -0.27$; $\chi^2(1) = 1.25$, $P = 0.263$, $n = 401$). Similar results are obtained in specification (2) that controls for performance, confidence, and age. Here we find that men actually decrease their competition entry rates in HIGH compared to NEUTRAL, the difference however being only marginally significant. The difference-in-differences, i.e., the impact of our priming intervention on the gender gap, is captured by the positive and significant interaction term female × HIGH. This confirms that priming with high power significantly influences the gender gap in competition entry, practically eliminating it (coefficient of female + female × HIGH $= 0.053$; $\chi^2(1) = 0.04$, $P = 0.842$ in column 2, $n = 401$). Further factors that influence entry rates significantly are ability (measured as the number of correctly solved calculations under the piece rate scheme), and confidence in one's competitive performance (measured by elicited beliefs).

**Mechanisms driving the effect of priming**. So what are the possible mechanisms that lead to the closing of the gender gap in the willingness to compete when participants are primed with high power? First, we can observe that participants in HIGH become much better calibrated in their competition entry decisions. Given that the rules of the competition in Stage 3 were such that one in four participants would win on average, well-calibrated decisions would require participants to enter the tournament at rates of 25% on average. This is the case in treatment HIGH, with rates statistically indistinguishable from 25% for both genders (two-sided binomial tests, $P = 0.394$, $n = 66$ for women; $P = 0.576$, $n = 68$ for men). On the contrary, entry rates in NEUTRAL are different from 25% for both genders, with men competing too much (40.28%; two-sided binomial test, $P = 0.004$, $n = 72$) and women competing too little (13.89%; two-sided binomial test, $P = 0.030$, $n = 72$). Besides, entry rates in LOW are different from 25% for men (with 38.1%; two-sided binomial test, $P = 0.020$, $n = 63$) but not for women (with 20%; two-sided binomial test $P = 0.456$, $n = 60$).

Another way of looking at the issue of well-calibrated entry decisions is to compare ex ante optimal entry decisions with actual entry rates. Actual winning rates in Stage 2 can be used as proxies for ex ante optimal entry decisions in Stage 3 (under the —satisfied—assumption that performance does not differ

significantly between Stage 2 and Stage 3, see Fig. 1): these rates are 28.3% for men and 21.7% for women. In HIGH, participants are indeed very well calibrated and entry rates are very close to optimal rates for men as well as for women (two-sided binomial tests, $P = 0.767$, $n = 66$ for women, $P = 1.000$, $n = 68$ for men). This is not the case for men in NEUTRAL (entry rate of 40.3% vs. optimal rate of 28.3%; two-sided binomial test, $P = 0.035$, $n = 72$) or LOW (entry rate of 38.1% vs. optimal rate of 28.3%; two-sided binomial test, $P = 0.093$, $n = 63$). Women's decisions in NEUTRAL and in LOW are better calibrated (two-sided binomial test, $P = 0.117$, $n = 72$; $P = 0.876$, $n = 60$, respectively, comparing between actual and optimal entry rates in NEUTRAL and LOW), which is in line with the observation that the priming intervention closes the gender gap primarily by changing decisions among male participants. It should be acknowledged, however, that optimal competition entry rates as calculated here are only a noisy measure of truly optimal behavior, due to the fact that participants received no feedback about how other group members had performed in the first two stages of the experiment.

Second, our dataset allows us to identify a concrete mediating factor as a potential driver of the observed priming effects. This factor is risk aversion, which is a well-known determinant of willingness to compete[5]. Elicited risk attitudes in our experiment are likely to be influenced by the priming manipulation, however, and can therefore not be treated as independent of treatment. Indeed, Supplementary Table 5 reveals that men are taking significantly more risk in NEUTRAL than in HIGH (amount invested into the risky asset: 8.63 experimental currency units in HIGH vs. 9.57 in NEUTRAL; Mann–Whitney $U$; $P = 0.094$, $n = 140$), which is a likely explanatory factor for the reduction in competition entry rates among males in HIGH. The difference in risk attitudes between the neutral and high-power treatment is not significant for women (Mann–Whitney $U$, $P = 0.263$, $n = 138$). Interestingly, if one includes risk attitudes into the regressions explaining competition entry (see column 3 in Supplementary Table 4), the variable HIGH (representing the effect of the high-power prime among males) becomes insignificant and the female dummy diminishes in size and statistical significance, which supports the role of risk aversion in explaining gender differences in competitiveness and the reaction to priming.

## Discussion

We have found that priming with power closes the gender gap in competitiveness. Such an inexpensive tool has therefore the capacity to generate a gender-balanced pattern in competitive behavior. Contexts where the use of such priming-based tools would be applicable include, among others, the educational system (especially given the fact that gender differences in the willingness to compete have been shown to emerge early in life[7]), or vocational training programs as part of active labor market policies. Our findings call for more research into the implementation and impact evaluation of priming interventions in these areas in practice. Actually, the particular power priming tool used in our study has been shown to improve the performance of participants in job or college admission interviews[35]. As a consequence, priming may contribute to a more balanced representation of women in top-level positions, where competing for attractive jobs is a must. A better gender balance may even support the longevity of a company[36] and provide more female role models[37,38] for future cohorts that compete on the labor market.

We have examined potential mechanisms through which priming affects competition entry rates and the size of the gender gap. In our experiment women react to the priming intervention

differently from men: they slightly (but insignificantly) increase their willingness to compete, while men decrease theirs. We consider the latter an important finding, since the experimental literature on competitiveness has shown that men are typically too overconfident and thus enter competitions too often, for which they pay a price in terms of lower expected payoffs[1,5–9]. The fact that male entry rates in LOW and NEUTRAL are very close to each other reveals that the difference between the low and high prime condition is not due to men primed with low power trying to restore their ego through increasing competitiveness; instead, the main effect of priming is the lower tournament entry rates among men, triggered by our high-power priming intervention.

While we consider this point an important insight into the behavioral effects of power priming, it also reveals one limitation of our study related to the real-world applicability of this intervention: why would men—if they knew the effects of high-power priming—agree to it? While it is beyond the scope of our paper to answer this question conclusively, we would like to note that high-power priming leads to a substantially improved calibration of competition entry decisions of both men and women. In particular, it avoids a great deal of disappointment and frustration of men who entered—overoptimistically—the competition and then failed. This fraction, while about 13% of men in NEUTRAL, goes practically down to zero in HIGH. It is also noteworthy that the reduced competition entry rates of men in HIGH do not cost them a significant amount of money. Actually, we find that priming with power does not reduce average earnings compared to the neutral condition (see Supplementary Tables 6 and 7 for details). This means that men do not pay a price for being primed with power, making it less likely that they would resist such an intervention. Finally, from a company's perspective, it is important to note that our priming intervention may have worked through an increased risk aversion as a mediating factor. Given that companies can suffer tremendously from employees taking too many risks—as has been shown in the last financial crisis in the aftermath of Lehman Brothers' bankruptcy[39]—an intervention that avoids excessive risk taking might be in the interest of companies.

While we have studied a key factor for success on labor markets, our study raises the interesting question whether power priming or other forms of priming[40] may also have systematic effects on other important economic preferences, such as time preferences[41] or social preferences[42]. Since these other domains of economic preferences have been shown to be influential for the long-run prospects of humans[43,44], we hope that future research will shed more light on how priming can also affect other important domains of economic decision making.

## Methods

**Experimental setup**. Using H-Root[45] and ORSEE[46], we recruited 424 experimental participants (212 men, 212 women) from the University of Innsbruck and the Max Planck Institute for Research in Collective Goods in Bonn, with an average age of 23.44 years (S.D. = 5.52; mean age by treatment equals 23.0 years in NEUTRAL, 23.8 years in LOW and 23.5 years in HIGH). They received an average payment of €14.68. All participants were undergraduate students of various academic fields, including business and economics (28.7% of the sample), natural sciences (14.7%), psychology (9.5%), education (5.2%), political sciences and sociology (4.5%), and other disciplines (37.4%).

Upon arrival subjects were randomly allocated to seats and received written instructions explaining the experimental tasks. The experimental instructions are reported in Supplementary Methods.

We drop 23 participants from the analysis because in a debriefing questionnaire they indicated suspicion of being primed and mentioned that the priming had influenced their decisions. Thus, we analyze data from 401 participants (see Supplementary Table 3 on gender composition).

**Code availability**. The computer code used in the experiment is available from the corresponding author on request.

## Data availability

The data that support the findings of this study are available from the corresponding author upon request.

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

## Acknowledgements

We thank Wilhelm Hoffmann for comments and Sarah Stuefer for research assistance. Financial support from the Universities of Innsbruck and Cologne and from the Max Planck Society is gratefully acknowledged.

## Author contributions

All authors have contributed equally to this study and share the responsibility for its content.

## Additional information

**Competing interests:** The authors declare no competing interests.

