## [Peer Review File · Nature Communications]

Reviewers' comments:

Reviewer #1 (Remarks to the Author):

Referee Report for: Closing the Gender Gap in Competitiveness Through Priming

This is a really good paper on the effect that psychological priming has on behavior. Dozens of laboratory experiments have shown that women often shy away from competitive environments with higher frequencies than men, and such findings have been advanced as a possible explanation for a portion of the gender-gap in salaries and promotions.

The authors propose to use psychological power priming to investigate whether the memory of "feeling of being in control" could help reduce the gender gap in competitiveness in a subsequent laboratory game. Their results show that priming can indeed significantly affect behavior and reduce to insignificant the initial gender gap.

These are my main remarks:

1. With respect to the Design, the authors use well-known instruments that have been previously validated. One departure from the classic game is that they payoffs change from Stage 2 ("tournament") to Stage 3 ("choice"). Under "tournament", the member that solved the highest number of calculations was paid euro 4 per correct answer while under "choice" the winner received only euro 2, not 4. Since the alternative choice to competition was to be paid a fixed amount, euro 1 per correct answer, could this perceived change in payoff have had an interacting influence with the induced priming? I.e. male subjects in HPP felt the effect of a perceived decrease in rewards from winning tournament more intensely than those in LPP, and hence reduced their competitiveness? This would point to a different mechanism for closing the gender gap.

2. With respect to the Results, the validity check displayed in Supplementary Table 1 shows that priming worked as expected, as observed with a higher stated value on the reported power scale for both men and women. Would it be possible to add in the supplementary appendix a balance check on the socioeconomic variables and mood questionnaire answers among the 4 treatment conditions reported in Supplementary Table 2? Especially because mood enters significantly in Supplementary Table 3 results.

3. Was the mood elicitation questionnaire administered before or after the power priming? If after priming, it is reasonable to expect mood to have been affected by that initial exercise too, so adding those three mood variables (AT, GB, CN) into the regressions in Supplementary Tables 1 (power) and 3 (competition) would cause issues with endogeneity.

4. I find surprising that in Table 3 belief/confidence as measured by question about Stage 2 display a negative and significant coefficient? I would have expected it positive, i.e., higher confidence about ability, higher competitiveness (as shown in the literature). Even a brief note would help the reader.

5. Interestingly, in Supplementary Table 3, power scale is insignificant. If priming works by increasing feeling of power/being in control, as shown by the validation exercise in Supplementary Table 1, shouldn't it have entered positively, significantly, making the significance on HPP vanish?

6. Why priming would affect positively (but insignificantly) women but negatively and significantly men? Men in HPP are shown to lower their desire to compete (model 2 and 3 in supplementary Table 3). Choice closer to "optimal", but the ex-ante optimal rates are calculated from actual winning rates in Stage 2, after priming already took place. 27.8% optimal is really close to 25% random. Maybe a different angle could be that instead of advancing that high-power priming make

men less competitive, low-power priming makes men enter competitive environment above optimal/random rates... to restore ego?

7. Please discuss a bit more on the concept that "Such interventions (quotas) have been found to cause unintended side-effects". For example, in a previous work, the authors (Balafoutas and Sutter) shows that quotas don't reduce efficiency. Here, the negative side effects are of a different kind: backfire. All policies have some drawbacks, we don't live in first-best worlds, so dismissing quotas like they do seem unnecessary at this point. Backfire may be only temporary, priming may have negative side-effects too, so in the motivation I would say "complementary, less intrusive ways..." (Page 3 line 56), instead of "alternative...".

This brings to an important question: how to actually implement such priming in situations outside the laboratory? Quotas are, at least conceptually, straightforward to implement. In the discussion the authors state it is "easy" to implement priming in actual labor markets, but many jobs/positions are all about taking competitive decision for the firm/business. Stated differently, the main issue does not seem to be only the one-time decision to apply or not for a CEO position, but rather it should reflect also the competitive decisions that such position require to take on a daily basis as part of its job description.

Reviewer #2 (Remarks to the Author):

This paper presents an experiment aiming to eliminate the fairly well established sex difference in competitiveness. It combines a tool from psychology--power priming--with an established competition paradigm from behavioral economics. Results indicate that when participants are in the high power prime (HPP) condition, the sex difference in competitiveness is no longer significant, which is not true in the low power prime (LPP) condition, or in the vast majority of previously published studies using this competition paradigm (without primes).

This paper has many strengths: (1) it is creative in its methods, combining psychology and behavioral economics; (2) it is elegantly and concisely written, easy to understand, and well researched; (3) the figures are effective; (4) the topic has practical significance and will be of interest to many scientists and citizens.

The paper also has substantial limitations and weaknesses. One major weakness is that the results don't fit the hypothesis developed in the Introduction (line 65-67). According to the hypothesis, being primed to feel powerful (HPP) should make individuals, especially women, more likely to compete. However, the results indicate that the high power prime did not affect women's competitiveness. The sex difference was reduced because men decreased their competitiveness in HPP condition. The authors do not anticipate this effect or even attempt to explain it. I think this is a major lapse. (I'll note that the authors crafted their Abstract to almost disguise the fact that the sex difference is completely due to the men becoming less competitive in HPP condition, whereas women were apparently unaffected by the primes. This shows the authors' ingenuity but it almost strikes me as dishonest.)

My intuition is that men (but not women!) who are primed to feel low in power (asked to recall when someone exerted control over them) become agitated and competitive to try to remove themselves from this situation (or the subjective feelings typically associated with this kind of situation) by competing as soon as they have the chance. Additional data and a neutral condition (which this experiment lacked) might be helpful to address if this kind of interpretation is correct.

A second weakness of this paper is that we really don't know how robust the results are. I don't have any particular suspicion about this experiment, but, in the wake of the so-called replication

crisis, I think it's perfectly appropriate to be skeptical. The replication crisis refers to the repeated documentation that many (perhaps most) published findings in many areas of science are not reliable. This replication problem has been discussed in psychology as much as in any other field, and priming studies have become particularly infamous for their unreliability. I have not heard of any specific problems with the power prime the authors use in the current study, but I still strongly believe that the experiment should be replicated to show that the initial results were not a fluke. I'd like the whole experiment to be replicated exactly (not conceptually). The only adjustment I'd suggest would be introducing a neutral condition (not HPP or LPP). I've seen more and more psychology articles including direct replications of their initial experiments, and I think these articles are far stronger than ones based on only a single experiment.

A third weakness of this paper is that, despite the authors' suggestions (beginning of Discussion section), there is little reason to believe that the experiment will have any real world or practical significance. Priming effects have become infamous for their unreliability (see above) but they are also criticized because, even when reliable, they seem to mainly (or only) work in laboratory situations. If we were to attempt to directly apply the current experiment to wage negotiation situation, then we would need give men, using deception, a high power prime that would reduce their competitiveness to women's typical levels. This would cause men not to negotiate as competitively so that they wouldn't earn as much as they otherwise could. There are some major ethical challenges here! Moreover, many things would be far different in the wage negotiation situation than in the current experimental situation. For example, the stakes would be far higher; individuals would have prepared for the negotiation; individuals would in a competitive interpersonal situation complete with dynamic social behavior of a living person (rather than the sterile experimental situation with anonymous potential competitors.) I realize that the vast majority of published behavioral effects do not readily translate to real-world situations, so I might be setting a high bar here. However, the authors seem to be making a claim that their work will translate, so I think it's fair to point out that this claim is very unlikely to be true.

Here are some specific things, many of which relate to points I raised above:

Line 72 "priming has been shown to successfully change how people think and behave." Please provide references for this claim. I don't doubt that some studies have shown priming reliably affects behavior, but I wonder if anyone has ever shown that priming is powerful enough to actually affect meaningful behavior in the real world ("external validity" in psychological parlance), rather than under experimental conditions. I'd like the authors to state that some priming effects have been shown to have ecological validity and to give evidence for this. Perhaps taking a sentence to provide a strong example (if one exists!) would be worthwhile. Perhaps this could be discussed more fully at the beginning of the Discussion (Line 200) where the authors are quite explicit in their claim that these results could have practical implications. Again, please provide an example or two make this claim more plausible.

Line 102. Presumably the participants in the study did not see or learn anything about their potential competitors. The supplemental materials also suggest this. Was this true? Please clarify in the main text.

Line 212 "both genders improve the calibration of competition entry choices as evinced by actual entry rates becoming indistinguishable from optimal entry rates under high-power priming." I don't really understand these claims about optimal entry rates because all the analyses are at the population level. It seems that the authors should examine individual decisions based on the individual's performance in Tasks 1 and 2 prior to the competition decision in Task 3. I suppose that making these analyses truly meaningful would require giving competitors feedback on how other people were performing, and this wasn't done (for reasonable reasons). My main point is that the extremely limited information that participants had in this experiment might not tell us much about how people make decisions under naturalistic conditions where they would have many

sources of information about themselves and their competitors. I'd like this to be acknowledged.

Line 80. Who are the participants? Undergraduates majoring in psychology or economics? Graduate students? How were they recruited? What are their ages? What are their work experiences? It's difficult to interpret the experiment without this kind of information.

Line 89 "...subjects reported significantly higher numbers on average in HPP (6.18) than in LPP (3.69; $z = -6.45$, $P = 0.000$, Wilcoxon-Mann-Whitney test..." Please provide the descriptive statistics separately for men and women. Do this here or in the Supplementary Tables.

Line 158 "Here we find that men actually decrease their competition entry rates significantly in HPP compared to LPP." So men power priming significantly decreases competitiveness in men, but I'm sure many readers (including this reviewer) would like to know if power priming significantly increases competitiveness in women. The slight difference (15.6 vs. 22.9) (and the fact that the authors didn't report this important result) suggests that power priming is ineffective for women. This would seem to undermine the implied interpretation of the experiment.

Responses to Referee #1

Referee comments are pasted below. Our point-by-point responses follow after each comment, in italics and indented.

This is a really good paper on the effect that psychological priming has on behavior. Dozens of laboratory experiments have shown that women often shy away from competitive environments with higher frequencies than men, and such findings have been advanced as a possible explanation for a portion of the gender-gap in salaries and promotions.

The authors propose to use psychological power priming to investigate whether the memory of “feeling of being in control” could help reduce the gender gap in competitiveness in a subsequent laboratory game. Their results show that priming can indeed significantly affect behavior and reduce to insignificant the initial gender gap.

Let us begin by thanking you for these kind words and for the favorable initial assessment of our work, as well as for the many useful comments. We have done our best to address all your concerns in this revised version of our manuscript, while also taking into account the comments by the editor and by the other referee. We should point out that, as requested by the other referee, we have collected additional data: we have doubled the sample size for the low and the high power prime condition, and in addition we have collected data for a neutral condition. In this neutral condition, participants were not primed and did not complete a writing task. As you can see in the updated Results section, the main insights of the paper remain robust, and the new Neutral condition allows to better assess the effects of priming and the differences between conditions.

These are my main remarks:

1. With respect to the Design, the authors use well-known instruments that have been previously validated. One departure from the classic game is that they payoffs change from Stage 2 (“tournament”) to Stage 3 (“choice”). Under “tournament”, the member that solved the highest number of calculations was paid euro 4 per correct answer while under “choice” the winner received only euro 2, not 4. Since the alternative choice to competition was to be paid a fixed amount, euro 1 per correct answer, could this perceived change in payoff have had an interacting influence with the induced priming? I.e. male subjects in HPP felt the effect of a perceived decrease in rewards from winning tournament more intensely than those in LPP, and hence reduced their competitiveness? This would point to a different mechanism for closing the gender gap.

It is true that the incentives in our paper differ with respect to most of the previous literature, since entering the competition provides on average a lower payoff compared to the piece rate. Motivated by the well documented phenomenon of participants (especially men) entering tournaments at extremely high rates (e.g., 73% in the 2007 paper by Niederle and Vesterlund), in Stage 3 we opted for an environment where tournament entry would occur at more “reasonable” frequencies. We did not see any reason why the difference in incentives between Stages 2 and 3 would interact with priming, although it is true that we cannot rule it out.

Perhaps one pattern in the new dataset that speaks against such an interaction is the fact that entry rates in the new NEUTRAL condition are practically identical with the LPP condition (now called LOW in the revision to make the label more intuitive), which would likely not be the case if such an interaction was in place.

Regarding the strength of incentives we wish to point out that, in general, it is quite often the case that tournaments pay less than piece rates on average, so that our incentives in Stage 3 can be a good representation of reality. There is a recent paper (under revision in the Journal of Economic Behavior and Organization, see reference below) by Ragan Petrie and Carmit Segal, which investigates the role of prizes in moderating the gender gap in competitiveness and considers tournaments that are on average less or more profitable than the piece rate. Two relevant findings from this paper are: (i) that the gender gap persists at incentives that are weaker than the ones used previously in the literature, which is good news for the comparability of our findings to existing literature; (ii) that cost-minimizing firms would choose the lowest tournament prizes possible, since these would be enough to attract good applicants. Especially this latter finding is important because it suggests that a setting with relatively low tournament prizes is likely to be observed in reality. We have not included this discussion in the manuscript, but we are happy to do so if you think it is important.

Reference: Petrie, R., Segal, C. Gender differences in competitiveness: The role of prizes.

Working Paper, available here:

http://www.raaganpetrie.org/uploads/8/4/4/3/84436206/gender_prizes_paper_march2017_final.pdf

2. With respect to the Results, the validity check displayed in Supplementary Table 1 shows that priming worked as expected, as observed with a higher stated value on the reported power scale for both men and women. Would it be possible to add in the supplementary appendix a balance check on the socioeconomic variables and mood questionnaire answers among the 4 treatment conditions reported in Supplementary Table 2? Especially because mood enters significantly in Supplementary Table 3 results.

We present below a balance check for the three mood dimensions, from which it becomes evident that mood is not evenly balanced across treatment. Specifically, the Good-Bad and the Active-Tired dimension differ significantly across treatments (p values referring to Mann-Whitney tests). This relates directly to your next comment below: you were right in guessing that mood is affected by priming, and therefore we no longer include it as an independent variable in the regressions (which is why we have not included the table below in the manuscript).

	NEUTRAL	LOW	HIGH	NEUTRAL vs. LOW	NEUTRAL vs. HIGH
Good-Bad Mood (GB)	4.01	3.04	2.84	p=0.011	p<0.001
Awake-Tired Mood (AT)	1.04	-0.37	0.13	p=0.002	p=0.033
Calm-Nervous Mood (CN)	3.20	3.15	3.39	p=0.853	p=0.512

Regarding age, it is nicely balanced across treatments: mean age equals 23.0 years in NEUTRAL, 23.8 in LPP, and 23.5 in HPP (now called HIGH in the revision). We mention this in the Methods section on page 12. The differences across treatments are insignificant, and the same is true if we additionally disaggregate by gender.

3. Was the mood elicitation questionnaire administered before or after the power priming? If after priming, it is reasonable to expect mood to have been affected by that initial exercise too, so adding those three mood variables (AT, GB, CN) into the regressions in Supplementary Tables 1 (power) and 3 (competition) would cause issues with endogeneity.

Thank you very much for this very insightful comment! Indeed, the mood elicitation questionnaire was administered after priming and we had missed this point. We have now run regressions of the three mood variables on low and high power priming (with Neutral as a baseline). Two of the three mood variables (AT and GB) are influenced by low as well as by high power priming, both treatment dummies being significant at least at the 5% level. The same is true when we regress the aggregate mood variable on the treatment dummies, with both dummies having a significant influence on mood. In terms of direction, HPP leads to a worse mood on average for both genders (differences between NEUTRAL and HPP in the Good-Bad dimension: $P = 0.057$ for men; $P = 0.002$ for women, Mann-Whitney U tests), and to a more tired state for women ($P = 0.040$). It is hard to think of a plausible story based on which these differences in mood can explain the pattern of treatment differences in entry rates across genders, hence we have simply dropped the mood variables from the analysis in order to avoid these issues of endogeneity.

However, your comment has led us to check for other variables that might have been affected by the priming tool, and we found out that priming had a strongly significant effect on our risk measure (also elicited after the priming). As it turns out, this effect can be useful in helping us offer an explanation for the observed treatment effects: men (but not women) become significantly more risk averse in HPP compared to NEUTRAL, which is a possible explanation for the puzzling reduction in male entry rates in HPP. While this is not conclusive evidence, it is nice that we have identified one concrete potential mechanism for this effect.

4. I find surprising that in Table 3 belief/confidence as measured by question about Stage 2 display a negative and significant coefficient? I would have expected it positive, i.e., higher confidence about ability, higher competitiveness (as shown in the literature). Even a brief note would help the reader.

We believe that this is just a small misunderstanding: as the notes to Table 3 explain, the belief variable is coded such that 1 corresponds to the most optimistic and 4 to the most pessimistic belief. Hence, in line with intuition, the negative coefficient means that higher confidence is associated with higher entry rates.

5. Interestingly, in Supplementary Table 3, power scale is insignificant. If priming works by increasing feeling of power/being in control, as shown by the validation exercise in Supplementary Table 1, shouldn't it have entered positively, significantly, making the significance on HPP vanish?

This is a very valid comment, and one that we can only speculate about. We find it plausible that the increase in feelings of power/being in control affects competition entry rates not directly, but through a further mediating mechanism, like for instance the ones we discuss in the paper (e.g., better calibration of entry rates). If this is true, then it is not unreasonable that the variable itself entered the regressions insignificantly. In any case, we note that this variable (power scale) is no longer included in the regressions, given that the variable is not available for participants in the neutral condition: the power scale refers to the particular situation from the writing task and it would be meaningless in NEUTRAL (where there is no writing task).

6. Why priming would affect positively (but insignificantly) women but negatively and significantly men? Men in HPP are shown to lower their desire to compete (model 2 and 3 in supplementary Table 3). Choice closer to "optimal", but the ex-ante optimal rates are calculated from actual winning rates in Stage 2, after priming already took place. 27.8% optimal is really close to 25% random. Maybe a different angle could be that instead of advancing that high-power priming make men less competitive, low-power priming makes men enter competitive environment above optimal/random rates... to restore ego?

Thank you for this very interesting suggestion. This is an explanation that we had not thought of previously, and indeed it seems plausible that men primed with low power would react by increasing competitiveness in order to restore their ego. This explanation was also proposed by the other referee. What speaks against such an explanation is the fact that entry rates for men are almost identical in the new neutral condition compared to LPP, and even a bit higher (40.3% vs. 38.1%); this explanation would imply that men enter the tournament more frequently in LPP compared to NEUTRAL, but they don't. Therefore, we have not included this potential explanation in the revised version of the paper.

7. Please discuss a bit more on the concept that "Such interventions (quotas) have been found to cause unintended side-effects". For example, in a previous work, the authors (Balafoutas and Sutter) shows that quotas don't reduce efficiency. Here, the negative side effects are of a different kind: backfire. All policies have some drawbacks, we don't live in first-best worlds, so dismissing quotas like they do seem unnecessary at this point. Backfire may be only temporary, priming may have negative side-effects too, so in the motivation I would say "complementary, less intrusive ways..." (Page 3, line 59), instead of "alternative...".

Thank you for this comment. Obviously, especially given our previous work, we never meant to dismiss quotas or other institutional approaches. We fully agree that all instruments can have side effects, and the point we were trying to make was that there is room and use for new tools and approaches (which might not be the case if for instance quotas had not been found to have any side effects at all). We have followed your suggestion and updated the relevant part of the text accordingly. We start the second paragraph of the paper as follows now: "For these

reasons, we consider it as important to examine complementary, less intrusive and inexpensive, ways to mitigate or even offset the gender differences in competitiveness.”

This brings to an important question: how to actually implement such priming in situations outside the laboratory? Quotas are, at least conceptually, straightforward to implement. In the discussion the authors state it is “easy” to implement priming in actual labor markets, but many jobs/positions are all about taking competitive decision for the firm/business. Stated differently, the main issue does not seem to be only the one-time decision to apply or not for a CEO position, but rather it should reflect also the competitive decisions that such position require to take on a daily basis as part of its job description.

Again, this is a very important issue and we thank you for motivating us to give it some more thought. It is true that the ways to implement a priming intervention are generally less clear than, say, implementing a quota policy. The other reviewer was also concerned about the issue of how to actually implement priming in the real world. We argue in the discussion section that one can think of ways to implement priming (in the educational system or as part of vocational training programs), and for this revision we have searched the literature for concrete evidence on the use of such interventions. We have come across a very useful study, in which experimental participants were primed using the exact same method as in our study, and then were evaluated by a panel of judges with regards to their performance in a job interview. These judges were students in one experiment, but experienced judges in a second experiment (professors who had the role of interviewer for entry applications into a business school). That study shows that participants primed with high power were more persuasive and successful in the interviews, compared to those in the neutral condition and those primed with low power. We view these findings as supportive of the real world relevance of the priming tool we are examining in our study. We now cite this study at the end of the first paragraph in the discussion section.

Reference: Lammers, J., Dubois, D., Rucker, D., Galinsky, A. (2013). Power gets the job: Priming power improves interview outcomes. Journal of Experimental Social Psychology 49, 776-779.

Responses to Referee #2

Referee comments are pasted below. Our point-by-point responses follow after each comment, in italics and a slight indentation.

This paper presents an experiment aiming to eliminate the fairly well established sex difference in competitiveness. It combines a tool from psychology--power priming--with an established competition paradigm from behavioral economics. Results indicate that when participants are in the high power prime (HPP) condition, the sex difference in competitiveness is no longer significant, which is not true in the low power prime (LPP) condition, or in the vast majority of previously published studies using this competition paradigm (without primes).

This paper has many strengths: (1) it is creative in its methods, combining psychology and behavioral economics; (2) it is elegantly and concisely written, easy to understand, and well researched; (3) the figures are effective; (4) the topic has practical significance and will be of interest to many scientists and citizens.

Let us begin by thanking you for these kind words and for the favorable initial assessment of our work, as well as for the many useful comments. We have done our best to address all your concerns in this revised version of our manuscript, while also taking into account the comments by the editor and by the other referee. As you will see in our detailed responses below, we have made extensive changes to the manuscript following your comments and we believe that this has helped us improve the paper to a large degree.

The paper also has substantial limitations and weaknesses. One major weakness is that the results don't fit the hypothesis developed in the Introduction (line 65-67). According to the hypothesis, being primed to feel powerful (HPP) should make individuals, especially women, more likely to compete. However, the results indicate that the high power prime did not affect women's competitiveness. The sex difference was reduced because men decreased their competitiveness in HPP condition. The authors do not anticipate this effect or even attempt to explain it. I think this is a major lapse. (I'll note that the authors crafted their Abstract to almost disguise the fact that the sex difference is completely due to the men becoming less competitive in HPP condition, whereas women were apparently unaffected by the primes. This shows the authors' ingenuity but it almost strikes me as dishonest.)

It is true that the high power condition worked in a different way than what we initially expected. Please see our thoughts on this issue as a response to your next comment, a few lines further down.

Regarding the comment that the absence of a difference was not sufficiently clear in the previous version, we kindly refer you to our response to your very last comment at the end of this letter; we hope that it is now entirely clear to the reader that the high power prime is successful in significantly reducing (and statistically eliminating) the gender gap, while the effects on each of two genders separately are not significant.

We would also like to note at this point that we were not able to identify a part in our abstract where we were trying to mislead the reader by suggesting that there is an effect of the high power prime for women, or by disguising anything: all we do say is that the prime closes down the gender gap. The relevant part was this one: "While in a low-power priming situation men are almost three times more likely than women to choose competition, this gap vanishes completely when subjects are primed with a high-power situation." We believe that this is an accurate statement, referring to the statistically significant difference-in-differences. Yet, in order to make it clearer that our intervention has an effect on men we conclude the new abstract with the following sentence: "We show that priming with high power makes competition entry decisions more realistic and also that it reduces the level of risk tolerance among male participants, which can help explain why it leads to a closing down of the gender gap in competitiveness."

My intuition is that men (but not women!) who are primed to feel low in power (asked to recall when someone exerted control over them) become agitated and competitive to try to remove themselves from this situation (or the subjective feelings typically associated with this kind of situation) by competing as soon as they have the chance. Additional data and a neutral condition (which this experiment lacked) might be helpful to address if this kind of interpretation is correct.

Thank you for sharing these thoughts. It is true that we were also surprised by the fact that the effect of the high prime condition works mostly through a reduction of male entry rates, rather than through an increase in female entry rates. We have invested considerable effort into coming up with potential explanations for this pattern, and the result of these efforts is primarily the analysis regarding the extent to which decisions are well calibrated, which can be found on pages 8-9.

The interpretation that men primed with low power become agitated and try to restore their ego by competing more is a very interesting one, and it was also suggested by the other referee. The new neutral condition – explained in detail in our response to your next comment – can help us test this explanation, and as it turns out it does not provide support for it. Entry rates for men are almost identical in the neutral condition compared to LPP (now called LOW to make the label more intuitive), and even a bit higher (40.3% vs. 38.1%): this explanation would imply that men enter the tournament more frequently in LPP compared to NEUTRAL but they don't. Therefore, we have not included this potential explanation in the revised version of the paper.

We note here that, following a comment by the other referee, we have checked for endogenous variables in the regressions and mood as well as risk turn out to be affected by treatment. While mood is not particularly helpful in explaining the observed treatment effects, risk can help us offer an additional explanation: men (but not women) become significantly more risk averse in HPP (now called HIGH in the revision) compared to NEUTRAL, which is a possible explanation for the puzzling reduction in male entry rates in HPP. While this is not conclusive evidence, it is nice that we have identified one concrete potential mechanism for this effect. This analysis is found on pages 9-10.

In any case, we do not claim that we are able to safely identify the exact mechanisms driving the observed treatment effects. In general, the difficulty in identifying such mechanisms is present in most priming studies, since priming can work in various ways that are hard to

control for. At the same time we should note that the main result (power prime closes the gender gap in competition entry rates) remains true irrespective of the precise way in which priming affects decisions.

A second weakness of this paper is that we really don't know how robust the results are. I don't have any particular suspicion about this experiment, but, in the wake of the so-called replication crisis, I think it's perfectly appropriate to be skeptical. The replication crisis refers to the repeated documentation that many (perhaps most) published findings in many areas of science are not reliable. This replication problem has been discussed in psychology as much as in any other field, and priming studies have become particularly infamous for their unreliability. I have not heard of any specific problems with the power prime the authors use in the current study, but I still strongly believe that the experiment should be replicated to show that the initial results were not a fluke. I'd like the whole experiment to be replicated exactly (not conceptually). The only adjustment I'd suggest would be introducing a neutral condition (not HPP or LPP). I've seen more and more psychology articles including direct replications of their initial experiments, and I think these articles are far stronger than ones based on only a single experiment.

We have followed your suggestion and have collected a large amount of additional data for this revised version of our paper, increasing the sample size by a factor of three, from N=144 to N=432. In particular:

(i) We collected an additional 144 observations in a new treatment, called Neutral. We spent some time considering alternative ways of implementing the neutral condition. For instance, we considered the possibility of asking participants to write about a typical day in their lives, or about their day so far, or the day before. The problem with this approach is that it leads to some loss of control given that some days may be associated with completely different states of mind than others. Also, comparability with the other two conditions would be limited given that such a script would not refer to a concrete situation. A third issue is that a policy recommendation based on such a control would be very hard to imagine. Based on these considerations and also on numerous discussions with colleagues in seminars, we opted for a neutral condition in which participants do not perform any writing task. While this entails a small asymmetry compared to the other two treatments, we believe that overall it is the cleanest option. We also note that it has been common in the literature, for instance in the paper by Lammers et al. (2013) that we mention in our response to your next comment.

(ii) We doubled the sample for the low and high power conditions as you requested, meaning that we collected an additional 72 observations in each treatment, leading to a total sample size of N=144 in each of the three treatments (before dropping a few observations from the analysis based on the debriefing questionnaire in LPP and HPP, like we were also doing in the previous version). The additional data were collected in the original location (Innsbruck), but also in Bonn, in order to further enhance the credibility of our findings by adding a second location (we note that including a location dummy in the regressions does not have any effect on the results, and the location dummy is completely insignificant).

As you can see by reading the updated Results section, all our main findings are robust and remain qualitatively unaffected, although naturally some small changes are documented. The general message from this new dataset is that we observe a large and significant gender gap in competitiveness in Neutral and in LOW, but no gap in HIGH (the difference-in-differences also

being significant in the regressions, as it was in the previous version). Again, the effect of the high prime works for the most part through a reduction in the competitiveness of males, while the effect for females is positive but insignificant. In general the NEUTRAL condition is strikingly similar to LOW, while HIGH leads to distinctly different patterns. This is very nice in our opinion, since it shows that intervening with a high power prime would make a difference compared to the status quo of no intervention.

A third weakness of this paper is that, despite the authors' suggestions (beginning of Discussion section), there is little reason to believe that the experiment will have any real world or practical significance. Priming effects have become infamous for their unreliability (see above) but they are also criticized because, even when reliable, they seem to mainly (or only) work in laboratory situations. If we were to attempt to directly apply the current experiment to wage negotiation situation, then we would need give men, using deception, a high power prime that would reduce their competitiveness to women's typical levels. This would cause men not to negotiate as competitively so that they wouldn't earn as much as they otherwise could. There are some major ethical challenges here! Moreover, many things would be far different in the wage negotiation situation than in the current experimental situation. For example, the stakes would be far higher; individuals would have prepared for the negotiation; individuals would in a competitive interpersonal situation complete with dynamic social behavior of a living person (rather than the sterile experimental situation with anonymous potential competitors.) I realize that the vast majority of published behavioral effects do not readily translate to real-world situations, so I might be setting a high bar here. However, the authors seem to be making a claim that their work will translate, so I think it's fair to point out that this claim is very unlikely to be true.

Thank you for raising all of these points. You are of course right in pointing out the fact that translating lab findings to real world situations is not always straightforward. This applies to most experimental work in the lab, but we can see that the concern may be more pressing in the case of priming. While we agree that these concerns must be kept in mind, we are more optimistic regarding the external relevance of our findings, mainly for the following reasons:

(i) We have dropped the claim that a priming intervention is easy to implement (as we did in the first sentence of the discussion section of the first submission – which has been revised now), but we argue in the discussion section that one can think of ways to implement priming (in the educational system or as part of vocational training programs). Nevertheless, following this comment of yours and also a comment by the other reviewer, we have searched the literature for concrete evidence on the use of such interventions. We have come across a very useful study, in which experimental participants were primed using the exact same method as in our study, and then were evaluated by a panel of judges with regards to their performance in a job interview. These judges were students in one experiment, but experienced judges in a second experiment (professors who had the role of interviewer for entry applications into a business school). That study shows that participants primed with high power were more persuasive and successful in the interviews, compared to those in the neutral condition and those primed with low power. We view these findings as supportive of the real world relevance of the priming tool we are examining in our study. Notice that, in these experiments, an interpersonal interaction is present. We now cite this study at the end of the first paragraph in the discussion section.

Reference: Lammers, J., Dubois, D., Rucker, D., Galinsky, A. (2013). Power gets the job: Priming power improves interview outcomes. Journal of Experimental Social Psychology 49, 776-779.

(ii) Regarding the stakes, it is not clear that on average that in reality the difference in payoffs between a tournament winner and a person who does not enter a tournament is more pronounced (in relative terms, of course) than in our experiment. In this regard, we now cite a recent paper (under revision in the Journal of Economic Behavior and Organization) by Ragan Petrie and Carmit Segal, which investigates the role of prizes in moderating the gender gap in competitiveness and considers tournaments that are on average less or more profitable than the piece rate. Two relevant findings from this paper are: (i) that the gender gap persists at incentives that are weaker than the ones used previously in the literature, which is good news for the comparability of our findings to existing literature; (ii) that cost-minimizing firms would choose the lowest tournament prizes possible, since these would be enough to attract good applicants. Especially this latter finding is important because it suggests that a setting with relatively low tournament prizes is likely to be observed in reality. We have not included this discussion in the manuscript, but we are happy to do so if you think it is important.

Reference: Petrie, R., Segal, C. Gender differences in competitiveness: The role of prizes.

Working Paper, available here:

http://www.raganpetrie.org/uploads/8/4/4/3/84436206/qender_prizes_paper_march2017_final.pdf

(iii) Regarding your argument that intervening by means of priming may present ethical challenges, we note that, similarly to previous work evaluating the effectiveness of interventions such as affirmative action, here we evaluate the effectiveness of a priming intervention without making value statements on whether the results of such an intervention are desirable. What we can say is that, provided the goal of the policy is to close down the gender gap, the examined intervention is a good way of doing it.

We hope that our arguments go some way in convincing you that the real world relevance of our findings is not that limited, at least when one makes the qualitative statement that power priming affects behavior and that it works toward closing the gender gap in competitiveness.

Here are some specific things, many of which relate to points I raised above:

Line 72 “priming has been shown to successfully change how people think and behave.” Please provide references for this claim. I don’t doubt that some studies have shown priming reliably affects behavior, but I wonder if anyone has ever shown that priming is powerful enough to actually affect meaningful behavior in the real world (“external validity” in psychological parlance), rather than under experimental conditions. I’d like the authors to state that some priming effects have been shown to have ecological validity and to give evidence for this. Perhaps taking a sentence to provide a strong example (if one exists!) would be worthwhile. Perhaps this could be discussed more fully at the beginning of the Discussion (Line 200) where the authors are quite explicit in their claim that these results could have practical implications. Again, please provide an example or two make this claim more plausible.

Here we would like to refer once again to the two experiments in the above paper by Lammers et al. (2013), which provides some support for the ecological validity of the exact same priming tool used in our study.

Line 102. Presumably the participants in the study did not see or learn anything about their potential competitors. The supplemental materials also suggest this. Was this true? Please clarify in the main text.

This is correct. Participants did not know anything about the potential competitors, with one exception: they know that their group consists of two men and two women, as stated in the experimental instructions. We have now added this information to the main text of the paper, in the description of the experimental design on page 5 (line 108) where we write: "Other than this gender composition, participants received no further information regarding their potential competitors."

Line 212 "both genders improve the calibration of competition entry choices as evinced by actual entry rates becoming indistinguishable from optimal entry rates under high-power priming." I don't really understand these claims about optimal entry rates because all the analyses are at the population level. It seems that the authors should examine individual decisions based on the individual's performance in Tasks 1 and 2 prior to the competition decision in Task 3. I suppose that making these analyses truly meaningful would require giving competitors feedback on how other people were performing, and this wasn't done (for reasonable reasons). My main point is that the extremely limited information that participants had in this experiment might not tell us much about how people make decisions under naturalistic conditions where they would have many sources of information about themselves and their competitors. I'd like this to be acknowledged.

You are right. We now acknowledge this limitation on page 9 (lines 208-211), where we write: "It should be acknowledged, however, that optimal competition entry rates as calculated here are only a noisy measure of truly optimal behavior, due to the fact that participants had no information about how other group members had performed in the first two stages of the experiment."

Line 80. Who are the participants? Undergraduates majoring in psychology or economics? Graduate students? How were they recruited? What are their ages? What are their work experiences? It's difficult to interpret the experiment without this kind of information.

All participants were undergraduate students of various academic backgrounds. We have the information regarding their field of study, which we now mention in the Methods section on page 12: The largest group were students in economics and business (28.7% of the sample), followed by natural sciences (14.7%), psychology (9.5%), education (5.2%), political sciences and sociology (4.5%), and other disciplines (37.4%).

Participants in Innsbruck were recruited using the software H-Root, which is the standard platform for recruiting experimental participants from a subject pool of more than 5,000 students of the university in total (please note that in the previous version we were mistakenly mentioning the old software, ORSEE). As a matter of fact, ORSEE was used to recruit experimental participants for the replication at the Max-Planck Institute in Bonn. This information is included in the Methods section on page 12.

We have information on the age of our sample, which is on average 23.44 years (with a standard deviation of 5.52). This information is also included in the Methods section. We note that we control for age in the regression analysis.

Unfortunately we have no information regarding the work experience of our participants. In Austria (where Innsbruck is located), about 6 out of 10 students, however, work either part-time or full-time during their studies at a university (sources are in German and available upon request). Hence, it can be expected that the majority of our Innsbruck participants have some (even regular) work experience.

Line 89 "...subjects reported significantly higher numbers on average in HPP (6.18) than in LPP (3.69; $z = -6.45$, $P = 0.000$, Wilcoxon-Mann-Whitney test..." Please provide the descriptive statistics separately for men and women. Do this here or in the Supplementary Tables.

We have added this information broken down by gender on page 5, lines 97-101, and in the process of disaggregating by gender we thought it would also be nice to report that there are no gender differences in the reported power scale in LPP or in HPP.

Line 158 "Here we find that men actually decrease their competition entry rates significantly in HPP compared to LPP." So men power priming significantly decreases competitiveness in men, but I'm sure many readers (including this reviewer) would like to know if power priming significantly increases competitiveness in women. The slight difference (15.6 vs. 22.9) (and the fact that the authors didn't report this important result) suggests that power priming is ineffective for women. This would seem to undermine the implied interpretation of the experiment.

We apologize for not explicitly mentioning that the effect of HPP on entry rates is not significant for women. Given that the differences in the entry rates as well as the joint coefficients (female + female x HPP) in the regressions were (and still are) rather small, the effect on women is in the expected direction but clearly not significant. In the previous version we mentioned the two significant effects in the regressions (for men and especially the difference-in-differences), and our silence on the insignificant effect was never meant to suggest that that effect was significant too. In any case, in this revised version we make it very clear quite early in the results section that this effect on its own is not significant, see page 7 (lines 158-162) where we write: "Although the decrease (increase) in male (female) entry rates in HPP compared to NEUTRAL are not significant ($\chi^2(1) = 2.36$, $P = 0.12$ for men; $\chi^2(1) = 0.84$, $P = 0.36$ for women), the resulting difference-in differences corresponding to the effect of the high power prime on the gender gap is significant, as the following regression analysis reveals."

Reviewers' comments:

Reviewer #1 (Remarks to the Author):

Please see attached .pdf report for detailed answers.

Second Referee Report for: Closing the Gender Gap in Competitiveness Through Priming

This revised version of the manuscript is much improved. I appreciate the newly added replication and addition of the new neutral treatment to better understand the mechanisms at play with priming. Here are my remarks on how the authors addressed my initial concerns.

1. With respect to the Design, the authors use well-known instruments that have been previously validated. One departure from the classic game is that they payoffs change from Stage 2 (“tournament”) to Stage 3 (“choice”). Under “tournament”, the member that solved the highest number of calculations was paid euro 4 per correct answer while under “choice” the winner received only euro 2, not 4. Since the alternative choice to competition was to be paid a fixed amount, euro 1 per correct answer, could this perceived change in payoff have had an interacting influence with the induced priming? I.e. male subjects in HPP felt the effect of a perceived decrease in rewards from winning tournament more intensely than those in LPP, and hence reduced their competitiveness? This would point to a different mechanism for closing the gender gap.

AUTHORS’ REPLY: It is true that the incentives in our paper differ with respect to most of the previous literature, since entering the competition provides on average a lower payoff compared to the piece rate. Motivated by the well documented phenomenon of participants (especially men) entering tournaments at extremely high rates (e.g., 73% in the 2007 paper by Niederle and Vesterlund), in Stage 3 we opted for an environment where tournament entry would occur at more “reasonable” frequencies. We did not see any reason why the difference in incentives between Stages 2 and 3 would interact with priming, although it is true that we cannot rule it out. Perhaps one pattern in the new dataset that speaks against such an interaction is the fact that entry rates in the new NEUTRAL condition are practically identical with the LPP condition (now called LOW in the revision to make the label more intuitive), which would likely not be the case if such an interaction was in place. Regarding the strength of incentives we wish to point out that, in general, it is quite often the case that tournaments pay less than piece rates on average, so that our incentives in Stage 3 can be a good representation of reality. There is a recent paper (under revision in the Journal of Economic Behavior and Organization, see reference below) by Ragan Petrie and Carmit Segal, which investigates the role of prizes in moderating the gender gap in competitiveness and considers tournaments that are on average less or more profitable than the piece rate. Two relevant findings from this paper are: (i) that the gender gap persists at incentives that are weaker than the ones used previously in the literature, which is good news for the comparability of our findings to existing literature; (ii) that cost-minimizing firms would choose the lowest tournament prizes possible, since these would be enough to attract good applicants. Especially this latter finding is important because it suggests that a setting with relatively low tournament prizes is likely to be observed in reality. We have not included this discussion in the manuscript, but we are happy to do so if you think it is important.

FURTHER COMMENT: I think it is very important to add this discussion in the main manuscript because it is both counterintuitive and contrary to what most economists would think: competitive situations offer higher rewards at the cost of facing higher risks. If competitive tournament would offer lower rewards than guaranteed piece-rate payment schemes, why would people ever enter uncertain competitive situations?

2. With respect to the Results, the validity check displayed in Supplementary Table 1 shows that priming worked as expected, as observed with a higher stated value on the reported power scale for both men and women. Would it be possible to add in the supplementary appendix a balance check on the socioeconomic variables and mood questionnaire answers among the 4 treatment conditions

reported in Supplementary Table 2? Especially because mood enters significantly in Supplementary Table 3 results.

AUTHORS' REPLY: We present below a balance check for the three mood dimensions, from which it becomes evident that mood is not evenly balanced across treatment. Specifically, the Good-Bad and the Active-Tired dimension differ significantly across treatments (p values referring to MannWhitney tests). This relates directly to your next comment below: you were right in guessing that mood is affected by priming, and therefore we no longer include it as an independent variable in the regressions (which is why we have not included the table below in the manuscript).

Regarding age, it is nicely balanced across treatments: mean age equals 23.0 years in NEUTRAL, 23.8 in LPP, and 23.5 in HPP (now called HIGH in the revision). We mention this in the Methods section on page 12. The differences across treatments are insignificant, and the same is true if we additionally disaggregate by gender.

FURTHER COMMENT: The authors should still report the balance check table in the supplementary material because it is interesting, even if some variables are not significantly different (which is good). Mood results are a bit puzzling. Why NEUTRAL condition is the one that scores the highest for Good-Bad mood?

3. Was the mood elicitation questionnaire administered before or after the power priming? If after priming, it is reasonable to expect mood to have been affected by that initial exercise too, so adding those three mood variables (AT, GB, CN) into the regressions in Supplementary Tables 1 (power) and 3 (competition) would cause issues with endogeneity.

AUTHORS' REPLY: Thank you very much for this very insightful comment! Indeed, the mood elicitation questionnaire was administered after priming and we had missed this point. We have now run regressions of the three mood variables on low and high power priming (with Neutral as a baseline). Two of the three mood variables (AT and GB) are influenced by low as well as by high power priming, both treatment dummies being significant at least at the 5% level. The same is true when we regress the aggregate mood variable on the treatment dummies, with both dummies having a significant influence on mood. In terms of direction, HPP leads to a worse mood on average for both genders (differences between NEUTRAL and HPP in the Good-Bad dimension: $P = 0.057$ for men; $P = 0.002$ for women, Mann-Whitney U tests), and to a more tired state for women ($P = 0.040$). It is hard to think of a plausible story based on which these differences in mood can explain the pattern of treatment differences in entry rates across genders, hence we have simply dropped the mood variables from the analysis in order to avoid these issues of endogeneity. However, your comment has led us to check for other variables that might have been affected by the priming tool, and we found out that priming had a strongly significant effect on our risk measure (also elicited after the priming). As it turns out, this effect can be useful in helping us offer an explanation for the observed treatment effects: men (but not women) become significantly more risk averse in HPP compared to NEUTRAL, which is a possible explanation for the puzzling reduction in male entry rates in HPP. While this is not conclusive evidence, it is nice that we have identified one concrete potential mechanism for this effect.

FURTHER COMMENT: Is mood now completely out of the story? I thought it was supposed to be part of the mechanism. In that case, even if with admittedly odd results, such analysis could stay in

the supplementary material because it may be of interest to other scientists working on the effect of priming.

4. I find surprising that in Table 3 belief/confidence as measured by question about Stage 2 display a negative and significant coefficient? I would have expected it positive, i.e., higher confidence about ability, higher competitiveness (as shown in the literature). Even a brief note would help the reader.

AUTHORS' REPLY: We believe that this is just a small misunderstanding: as the notes to Table 3 explain, the belief variable is coded such that 1 corresponds to the most optimistic and 4 to the most pessimistic belief. Hence, in line with intuition, the negative coefficient means that higher confidence is associated with higher entry rates.

FURTHER COMMENT: Good.

5. Interestingly, in Supplementary Table 3, power scale is insignificant. If priming works by increasing feeling of power/being in control, as shown by the validation exercise in Supplementary Table 1, shouldn't it have entered positively, significantly, making the significance on HPP vanish?

AUTHORS' REPLY: This is a very valid comment, and one that we can only speculate about. We find it plausible that the increase in feelings of power/being in control affects competition entry rates not directly, but through a further mediating mechanism, like for instance the ones we discuss in the paper (e.g., better calibration of entry rates). If this is true, then it is not unreasonable that the variable itself entered the regressions insignificantly. In any case, we note that this variable (power scale) is no longer included in the regressions, given that the variable is not available for participants in the neutral condition: the power scale refers to the particular situation from the writing task and it would be meaningless in NEUTRAL (where there is no writing task)

FURTHER COMMENT: Fine with leaving it out then.

6. Why priming would affect positively (but insignificantly) women but negatively and significantly men? Men in HPP are shown to lower their desire to compete (model 2 and 3 in supplementary Table 3). Choice closer to "optimal", but the ex-ante optimal rates are calculated from actual winning rates in Stage 2, after priming already took place. 27.8% optimal is really close to 25% random. Maybe a different angle could be that instead of advancing that high-power priming make men less competitive, low-power priming makes men enter competitive environment above optimal/random rates... to restore ego?

AUTHORS' REPLY: Thank you for this very interesting suggestion. This is an explanation that we had not thought of previously, and indeed it seems plausible that men primed with low power would react by increasing competitiveness in order to restore their ego. This explanation was also proposed by the other referee. What speaks against such an explanation is the fact that entry rates for men are almost identical in the new neutral condition compared to LPP, and even a bit higher (40.3% vs. 38.1%); this explanation would imply that men enter the tournament more frequently in LPP compared to NEUTRAL, but they don't. Therefore, we have not included this potential explanation in the revised version of the paper.

FURTHER COMMENT: I think a footnote stating precisely that, and why the data show it is not the case, would help the many readers who probably would come up with the same thought.

7. Please discuss a bit more on the concept that “Such interventions (quotas) have been found to cause unintended side-effects”. For example, in a previous work, the authors (Balafoutas and Sutter) shows that quotas don’t reduce efficiency. Here, the negative side effects are of a different kind: backfire. All policies have some drawbacks, we don’t live in first-best worlds, so dismissing quotas like they do seem unnecessary at this point. Backfire may be only temporary, priming may have negative side-effects too, so in the motivation I would say “complementary, less intrusive ways...” (Page 3 line 56), instead of “alternative...”.

This brings to an important question: how to actually implement such priming in situations outside the laboratory? Quotas are, at least conceptually, straightforward to implement. In the discussion the authors state it is “easy” to implement priming in actual labor markets, but many jobs/positions are all about taking competitive decision for the firm/business. Stated differently, the main issue does not seem to be only the one-time decision to apply or not for a CEO position, but rather it should reflect also the competitive decisions that such position require to take on a daily basis as part of its job description.

AUTHORS’ REPLY: Thank you for this comment. Obviously, especially given our previous work, we never meant to dismiss quotas or other institutional approaches. We fully agree that all instruments can have side effects, and the point we were trying to make was that there is room and use for new tools and approaches (which might not be the case if for instance quotas had not been found to have any side effects at all). We have followed your suggestion and updated the relevant part of the text accordingly. We start the second paragraph of the paper as follows now: “For these reasons, we consider it as important to examine complementary, less intrusive and inexpensive, ways to mitigate or even offset the gender differences in competitiveness.”

Again, this is a very important issue and we thank you for motivating us to give it some more thought. It is true that the ways to implement a priming intervention are generally less clear than, say, implementing a quota policy. The other reviewer was also concerned about the issue of how to actually implement priming in the real world. We argue in the discussion section that one can think of ways to implement priming (in the educational system or as part of vocational training programs), and for this revision we have searched the literature for concrete evidence on the use of such interventions. We have come across a very useful study, in which experimental participants were primed using the exact same method as in our study, and then were evaluated by a panel of judges with regards to their performance in a job interview. These judges were students in one experiment, but experienced judges in a second experiment (professors who had the role of interviewer for entry applications into a business school). That study shows that participants primed with high power were more persuasive and successful in the interviews, compared to those in the neutral condition and those primed with low power. We view these findings as supportive of the real world relevance of the priming tool we are examining in our study. We now cite this study at the end of the first paragraph in the discussion section.

FURTHER COMMENT: While that study is really interesting, I am still not convinced that this is a realistic tool to close the gender gap as I don’t think men, if they’d knew that the gender gap is closed by lowering their competitiveness, would do it. If competitiveness is a good trait, why would men agree to lower it and lower their income?

FINAL COMMENT: The authors addressed most of my initial remarks satisfactorily. What I think it is still the Achilles’ heel of this paper is the fact that the gender gap is closed by a policy that I find

really hard to implement in practice, one based on a gender (men) willing to go through an artificial procedure to lower something that it is in their interest to maintain high.
Priming is still important to study to understand better laboratory behavior.

Reviewer #2 (Remarks to the Author):

This manuscript has been substantially improved and it has addressed most of my major concerns.

I still think the interpretation of the main finding (men decrease competitiveness when primed for power) is problematic. I still have some skepticism regarding the practical implications of the study.

However, I have no major objections to this manuscript being published.

One minor note: Reference #25 was authored by Stapel, who has been convicted of scientific fraud and had more than 50 articles retracted. Please confirm that this paper has not been retracted. If it has been retracted, it should not be cited.

Responses to Referee 1

This manuscript has been substantially improved and it has addressed most of my major concerns. I still think the interpretation of the main finding (men decrease competitiveness when primed for power) is problematic. I still have some skepticism regarding the practical implications of the study. However, I have no major objections to this manuscript being published.

We are very glad to hear that you find the manuscript substantially improved, and that we were able to address your major concerns.

In the discussion section (penultimate paragraph of the discussion section on pages 11-12) we discuss the limitations of our intervention in practice. We acknowledge that the applicability of priming may be difficult, and highlight the need for more research into the implementation and impact evaluation of such interventions in reality. However, we also put forward three reasons why it may not be completely unrealistic to even win men over for such an intervention. These reasons are: (i) that high power priming makes entry decisions better calibrated, for which reason disappointment of men (who entered competition, but lost) can be avoided, (ii) that power priming has no significant effects on earnings, neither for women nor for men (supported through additional analysis in Supplementary Tables 6 and 7), and (iii) that companies may benefit from interventions that reduce the level of risk taking (in particular of men), as our intervention has been shown to be mediated by risk aversion. We hope that this extensive discussion satisfies you.

Please note that, in compliance with the author guidelines, changes and additions in the manuscript text file are highlighted in gray in this revised version.

One minor note: Reference #25 was authored by Stapel, who has been convicted of scientific fraud and had more than 50 articles retracted. Please confirm that this paper has not been retracted. If it has been retracted, it should not be cited.

Thank you for this comment and making us aware of this issue. We checked if the mentioned reference #25 (Lammers, J., Stapel, D. A., & Galinsky, A. D., Psychological Science, 2010) has been retracted, and fortunately this is not the case. According to a statement by Psychological Science, this specific article was among those that „have been cleared by the Levelt Committee, whose investigation into these articles found no evidence of fraudulent data or practices“. See also the following link: <http://journals.sagepub.com/doi/abs/10.1177/0956797612453137>

Responses to Referee 2

This revised version of the manuscript is much improved. I appreciate the newly added replication and addition of the new neutral treatment to better understand the mechanisms at play with priming. Here are my remarks on how the authors addressed my initial concerns.

We are very glad to hear that you find the manuscript much improved, and we agree that the new enlarged dataset has been helpful in terms of understanding the effects of priming. From the seven comments in your initial report, you have included further remarks on five (comments no. 1, 2, 3, 6 and 7). Below we paste those further remarks from your latest report, along with our detailed responses. Thank you for these additional comments, which we have found very useful.

Please note that, in compliance with the author guidelines, changes and additions in the manuscript text file are highlighted in gray in this revised version.

1. FURTHER COMMENT: I think it is very important to add this discussion in the main manuscript because it is both counterintuitive and contrary to what most economists would think: competitive situations offer higher rewards at the cost of facing higher risks. If competitive tournament would offer lower rewards than guaranteed piece-rate payment schemes, why would people ever enter uncertain competitive situations?

We have now added this discussion to the main manuscript, in the middle paragraph of page 6.

Regarding your question of why people would ever prefer uncertain competitive situations that pay less than piece rates on average, we believe that a large part of the explanation for this sort of behavior relates to overconfidence. (Over)confidence has been shown to play a major role in tournament entry decisions in the literature, and this is the case also in our paper (as seen in the highly significant coefficients on beliefs about competitive performance in Supplementary Table 4). For participants who assign a high enough probability to winning the tournament, competing is still preferable to the piece rate, even though it may in reality be suboptimal (see the nice paper by Larkin and Leider, 2012, cited as reference 32 now, on how overconfident employees forgo higher payoffs by sorting too often into non-linear incentive schemes). In the discussion section, on page 11, lines 262-265 we come back to this issue by stating that “the experimental literature on competitiveness has shown that men are typically too overconfident and thus enter competitions too often, for which they pay a price in terms of lower expected payoffs”.

One further possible explanation for competition entry despite lower expected earnings relates to a utility that some participants may gain from the mere fact that they are competing, as well as to a large utility that can be gained from winning the tournament, and which can extend beyond monetary payoffs.

2. FURTHER COMMENT: The authors should still report the balance check table in the supplementary material because it is interesting, even if some variables are not significantly different (which is good). Mood results are a bit puzzling. Why NEUTRAL condition is the one that scores the highest for Good-Bad mood?

We have now added the mood balance check to the SI, as Supplementary Table 2. Regarding the fact that mood is better in NEUTRAL compared to LOW and HIGH, we can only speculate

about possible reasons. One such reason might be concerns for equality and fairness that might worsen a subject's mood: both priming conditions ask participants to write about situations that feature an asymmetric distribution of power, and therefore it is not necessarily surprising that this may create adverse psychological effects.

3. FURTHER COMMENT: Is mood now completely out of the story? I thought it was supposed to be part of the mechanism. In that case, even if with admittedly odd results, such analysis could stay in the supplementary material because it may be of interest to other scientists working on the effect of priming.

Mood was included among the explanatory variables in the regressions of the original version of the paper, but has now been removed from the analysis due to the very important endogeneity issue that you raised in your previous report. As described in our response to your comment 2 above, the new Supplementary Table 2 includes the information that mood is not independent of treatment. In the notes to this table, the reader can now find detailed information on how mood varies by treatment.

6. FURTHER COMMENT: I think a footnote stating precisely that, and why the data show it is not the case, would help the many readers who probably would come up with the same thought.

Unfortunately we cannot add a footnote, since the author guidelines of the journal state that footnotes are not used. We have looked for an appropriate place in the text where we can add a short discussion on this point, and we decided to add the following text in the discussion section (lines 265-269 on page 11): "The fact that male entry rates in LOW and NEUTRAL are very close to each other reveals that the difference between the low and high prime condition is not due to men primed with low power trying to restore their ego through increasing competitiveness; instead, the main effect of priming is the lower tournament entry rates among men, triggered by our high power priming intervention".

7. FURTHER COMMENT: While that study is really interesting, I am still not convinced that this is a realistic tool to close the gender gap as I don't think men, if they'd knew that the gender gap is closed by lowering their competitiveness, would do it. If competitiveness is a good trait, why would men agree to lower it and lower their income?

FINAL COMMENT: The authors addressed most of my initial remarks satisfactorily. What I think it is still the Achilles' heel of this paper is the fact that the gender gap is closed by a policy that I find really hard to implement in practice, one based on a gender (men) willing to go through an artificial procedure to lower something that it is in their interest to maintain high. Priming is still important to study to understand better laboratory behavior.

We reply to these last two comments together, since they relate to the reviewer's concern that men would not embrace the proposed policy. This is an important issue that must be acknowledged and discussed. We would like to offer the following thoughts on the issue:

It is important to keep in mind that competing is not necessarily a good thing, as we have also argued in our response to your very first comment above: a participant would be well advised to compete if he/she would go on to win the tournament, but he should not compete if he would lose the tournament. We are already showing in the paper that the high priming condition leads to a better calibration of decisions compared to the neutral condition, suggesting that it helps

people make better decisions. This way, our priming intervention might even avoid disappointment and frustration among men who choose competition, but then lose. Recall that in NEUTRAL there is a gap of about 13 percentage points between the fraction of men entering competition and the fraction of men actually winning. While entering competition might create a utility from the excitement and the prospects of winning, losing the competition may as well create disappointment and frustration, not the least because it leads to zero payoffs.

To offer some concrete evidence on whether priming is unattractive for men because entering competition less often reduces their payoffs, we present below two additional pieces of statistical analysis. First, in Table R1 we present the earnings in Stage 3 by gender and priming condition. We find that earnings for men are only slightly higher in NEUTRAL compared to HIGH, but the difference is entirely insignificant ($p=0.92$, Mann-Whitney test). In the regressions of Table R2 we control for Stage 3 performance, and the coefficient on HIGH is now even slightly positive, and still insignificant. All treatment dummies are insignificant for both genders and the only significant predictor of Stage 3 earnings is performance. Hence, on average, priming with high power does not reduce men's earnings in the affected Stage 3, which means that applying it in practice may not be so controversial for men.

We have included this discussion in the penultimate paragraph of the discussion section (see in particular lines 273-282 on pages 11-12,) and have added the additional data analysis to the SI, as Supplementary Tables 6 and 7. In the same paragraph (lines 270-273) we also acknowledge the limitations of our study with respect to applicability, and hope that you deem our discussion satisfactory.

Table R1: Average earnings (in €) in Stage 3, by gender and priming condition ($N = 401$). Standard deviations in parentheses.

	Men	Women	Both
NEUTRAL	7.486 (7.517)	6.444 (4.155)	6.965 (6.074)
LOW	6.016 (5.726)	6.683 (6.179)	6.341 (5.937)
HIGH	7.044 (6.391)	5.772 (4.607)	6.418 (5.598)

Table R2: Regression results on earnings in Stage 3, by gender

	Men	Women
LOW	-0.31 (0.93)	0.74 (0.65)
HIGH	0.39 (0.91)	-0.05 (0.63)
performance Stage 3	1.28*** (0.12)	1.24*** (0.10)
constant	-2.66** (1.16)	-2.25*** (0.81)
N	203	198

Notes: Dependent variables: Earnings in Stage 3. Ordinary least squares regressions. Standard errors in parentheses. * $p < 0.10$, ** $p < 0.05$, *** $p < 0.01$

REVIEWERS' COMMENTS:

Reviewer #1 (Remarks to the Author):

I find the new analysis performed to tests whether or not men would be worse off from the priming intervention satisfying. So, if priming with high power would not reduce men's earnings, applying it in practice may indeed not be so controversial for men after all.

I have no further comments. The current version of this paper addresses all the questions I had and I think it offers a good contribution to the literature.